# Neuromodulation Using Transcranial Focused Ultrasound on the Bilateral Medial Prefrontal Cortex

**DOI:** 10.3390/jcm11133809

**Published:** 2022-06-30

**Authors:** Young Goo Kim, Song E. Kim, Jihye Lee, Sungeun Hwang, Seung-Schik Yoo, Hyang Woon Lee

**Affiliations:** 1Departments of Neurosurgery, Ewha Womans University Mokdong Hospital, Seoul 07985, Korea; ygkim@ewha.ac.kr; 2Departments of Neurology, Ewha Womans University School of Medicine and Ewha Medical Research Institute, Seoul 07985, Korea; vjf-i@ewha.ac.kr (S.E.K.); pairas@naver.com (J.L.); neurosung@gmail.com (S.H.); 3Departments of Medical Science, Ewha Womans University School of Medicine and Ewha Medical Research Institute, Seoul 07985, Korea; 4Computational Medicine, Graduate Program in System Health Science & Engineering, Ewha Womans University, Seoul 03765, Korea; 5Department of Radiology, Brigham and Women’s Hospital, Harvard Medical School, Boston, MA 02115, USA; yoo@bwh.harvard.edu

**Keywords:** transcranial focused ultrasound, medial prefrontal cortex, sonication parameters, excitatory, suppressive, modulation

## Abstract

Transcranial focused ultrasound (tFUS) is a promising technique of non-invasive brain stimulation for modulating neuronal activity with high spatial specificity. The medial prefrontal cortex (mPFC) has been proposed as a potential target for neuromodulation to prove emotional and sleep qualities. We aim to set up an appropriate clinical protocol for investigating the effects of tFUS stimulation of the bilateral mPFC for modulating the function of the brain-wide network using different sonication parameters. Seven participants received 20 min of 250 kHz tFUS to the bilateral mPFC with excitatory (70% duty cycle with sonication interval at 5 s) or suppressive (5% duty cycle with no interval) sonication protocols, which were compared to a sham condition. By placing the cigar-shaped sonication focus on the falx between both mPFCs, it was possible to simultaneously stimulate the bilateral mPFCs. Brain activity was analyzed using continuous electroencephalographic (EEG) recording during, before, and after tFUS. We investigated whether tFUS stimulation under the different conditions could lead to distinctive changes in brain activity in local brain regions where tFUS was directly delivered, and also in adjacent or remote brain areas that were not directly stimulated. This kind of study setting suggests that dynamic changes in brain cortical responses can occur within short periods of time, and that the distribution of these responses may differ depending on local brain states and functional brain architecture at the time of tFUS administration, or perhaps, at least temporarily, beyond the stimulation time. If so, tFUS could be useful for temporarily modifying regional brain activity, modulating functional connectivity, or reorganizing brain functions associated with various neuropsychiatric diseases, such as insomnia and depression.

## 1. Introduction

Transcranial focused ultrasound (tFUS) is a promising method of non-invasive brain stimulation to modulate neuronal activity with high spatial specificity compared to other non-invasive neuromodulation tools, such as transcranial magnetic stimulation (TMS) and transcranial direct current stimulation (tDCS) [1]. Low-intensity ultrasound can be transmitted through the skull and reach deep targets in the brain [2,3]. The efficacy and safety of neuromodulation using tFUS has been demonstrated in animals [4,5]. In humans, neuromodulation with tFUS has been demonstrated in the somatosensory [6,7], primary motor cortex [8], visual cortex [9], and thalamus [10,11]. In addition, the possibility of the therapeutic application of tFUS has been shown in several studies reporting the effects of FUS stimulation on patients with chronic pain [12], post-traumatic disorder of consciousness [13], and Alzheimer’s disease [14,15]. 

For ultrasound stimulation to produce neuromodulation effects, it is important to choose appropriate stimulation parameters. Non-invasive tFUS can have excitatory or suppressive effects on brain activity by manipulating ultrasound stimulation parameters, including the acoustic frequency, amplitude, duration, duty cycle, and pulse repetition frequency [16,17]. Animal and human studies have shown that tFUS with various parameters modulates excitatory or suppressive frequency [16,17]. However, previous studies showed conflicting results due to their different ultrasound parameters and experimental conditions. For clinical and therapeutic applications of tFUS, it is important to clarify the ultrasound stimulation parameters causing excitatory and suppressive effects on brain activity [5,17].

The medial prefrontal cortex (mPFC) plays an essential role in cognitive control and emotional regulation. In addition, the mPFC is a primary region of the resting-state brain network known as the default mode network (DMN), which is mainly involved in consciousness [18], sleep [19], cognition [20], and mood [21]. The stimulation of the mPFC may alter metabolic activity and neural connections associated with this region and modulate connected brain networks [22]. In fact, the mPFC has been suggested as a potential target for brain activity and network changes for neuromodulation, such as improving cognitive function and treating behavioral and emotional disorders [23]. Notably, the mPFC is centrally located deep in the frontal lobe, and it is difficult to target stimuli using other neuromodulatory tools limited mainly to the cortical surface [24]. In addition, since tFUS can precisely determine the stimulation target based on brain structural MRI, tFUS has an advantage over other non-invasive neuromodulation tools that target the prefrontal on the scalp.

To the best of our knowledge, the effects of the inhibition or excitation of the mPFC using tFUS on brain activity have been rarely studied. This study aims to determine whether tFUS on the bilateral mPFC with different parameters exhibits modulating effects on brain activity recorded by electroencephalography (EEG). The effects of tFUS stimulation on the mPFC were tested under three different conditions with excitatory, suppressive, and sham effects. Changes in brain activity were investigated by recording brain neuronal activity before and after tFUS with different parameters. For this reason, we aim to validate the experimental setup and ultrasound stimulation parameters in this study. The study design, brain targeting, tFUS procedures, data acquisition, and analysis from EEG are described in great detail in the following paper.

## 2. Materials and Methods

### 2.1. Recruitment

Right-handed healthy volunteers aged from 45 to 65 years were recruited for this study. The exclusion criteria were as follows: (1) history of neurological conditions including stroke, cerebral edema, tumor, or epilepsy; (2) psychiatric disorders such as anxiety or major depressive disorder; (3) significant cognitive or communicative impairments; (4) having incompatibility with MRI, i.e., a metallic implant in the brain or claustrophobia; (5) in baseline neuroimage (CT and structural MRI), extensive calcifications or extended frontal sinus being in the sonication pathway that cannot be avoided by tailoring the treatment plan; (6) severe parenchymal atrophic change that cannot fully cover the bilateral mPFC by midline targeting; (7) more than 30% of the scalp or forehead skin in the sonication pathway covered by scars, skin disorders (e.g., eczema), or atrophy; and (8) pregnant or breastfeeding women. This study was approved by the Institutional Review Board of Ewha Womans University Mokdong Hospital and registered with the Clinical Research Information Service (CRIS) of the Ministry of Health and Welfare, Republic of Korea (approval No. EUMC 2019-12-013). All participants provided written informed consent approved by the Ethics Committee.

### 2.2. Study Design

This study used a randomized controlled trial design, allocating participants to three stimulation conditions: excitatory, suppressive, or sham stimulation. The study design and procedures are described in Figure 1. Each participant was required to visit the laboratory on three separate occasions to complete the experiments. Participants were first expected to go through a screening and baseline visit, reviewed for inclusion/exclusion criteria, and provided written informed consent. Baseline neuroimaging, EEG, and self-reported questionnaires for demographic, health, and medical information were performed. During the second visit, one week after the screening and baseline visit, each participant was randomly assigned to one of three stimulation modes. One week after sonication on the second visit, all participants completed a third follow-up visit for assessment of side effects and safety.

### 2.3. Neuroimage Acquisition

Prior to the baseline neuroimage data acquisition, donut-shaped, self-adhesive fiducial markers (PinPoint; Beekley Corp., Bristol, CT, USA) were placed onto four locations on each subject’s head. These fiducial markers were identified in both MRI and CT scans, and their spatially distributed locations were used for co-registration between the subject’s actual head anatomy and the individual neuroimage during the sonication experiment. As mentioned above, all participants underwent CT scans to obtain information on the skull structure, such as extensive calcifications or extended frontal sinus within the sonication pathway with the following parameters: axial orientation, slice thickness 1.2 mm, field-of-view (FOV) 210 × 210 mm^2^, image matrix 512 × 512.

MRI was conducted to obtain anatomical and functional brain images using a 3 T MRI scanner (Philips, Best, The Netherlands) during baseline sessions. Anatomical T1-weighted images were acquired in the sagittal orientation with the following parameters: voxel size = 0.50 mm × 0.50 mm × 1.00 mm, matrix size = 448 × 448, and slice number = 160.

### 2.4. Low-Intensity Focused Ultrasound Procedure

We used an image-guided tFUS system (NS-US100, Neurosona Co., Ltd., Seoul, Korea) for tFUS of the mPFC. The detailed procedure of tFUS has been described previously [6,15]. Subject-specific co-registered CT and T1-weighted volumetric MRI were loaded on an image-guided ultrasound navigation program. The fiducial markers were placed on each participant’s head at four locations: behind the bilateral ears and above the bilateral eyebrows. The fiducial markers were used to spatially co-register the subject’s head and the corresponding anatomical neuroimage on the navigation system. We used a single-element, air-backed, spherical ultrasound transducer with an 80 mm diameter and 68 mm focal length. The spatial acoustic intensity profile generated by the tFUS transducer in this study showed that the size of the tFUS focus at full width with half-maximum intensity was approximately 9 mm in diameter and 50 mm in length (Figure 2).

Since we used an image-guided tFUS system using a single transducer, it was not possible to change the size of the sonication focus on an individual subject basis. However, the depth of focus was adjusted using compressible PVA hydrogels. Therefore, for effective sonication, subjects were initially excluded who had extensive calcification or dilated frontal sinus in the sonication pathway using baseline neuroimaging, including thin-slice CT and T1-MRI. In addition, as shown in Figure 3B, through pre-stimulation simulation using the baseline neuroimaging, it was confirmed that proper stimulation of the bilateral mPFC was possible by targeting the anterior falx. The center of the region of maximum intensity is 70 mm from the exit plane of the transducer on the sonication trajectory. The transducer was placed on the subject’s scalp with a compressible cryogel for acoustic coupling. A motion-tracking infrared camera detected the optical trackers attached to the transducer and headgear in real time. Three-dimensional image guidance based on spatially co-registered multi-modal neuroimaging allowed the operator to manually navigate the location of focus in the subject’s neuroanatomical space. The bilateral mPFCs as tFUS target location were identified in standard MNI space and inversely transformed to the participant’s original brain space. Sonication parameters were as follows: 20 min of 250 kHz tFUS to mPFC with excitatory (70% duty cycle with sonication interval at 5 s) or suppressive (5% duty cycle with no interval) stimulation protocols. With the acoustic intensity at a focus (AIF) of 3 W/cm^2^, the fundamental frequency was 250 kHz; with the acoustic intensity (AI) at 0.30 MPa, the mechanical index (MI) was 0.60 at the transducer without attenuation; 0.18 MPa, MI 0.36 (attenuation 40%); or 0.15 MPa, MI 0.30 (attenuation 50%) at the focal target for both excitatory and suppressive stimulations.

### 2.5. Bilateral Medial Prefrontal Cortex Targeting

Anatomical MRI-based targeting was conducted using statistical parametric mapping 8 (SPM8). SPM analysis is an alternative voxel-to-voxel analysis method that avoids subjectivity. To determine stimulation target points on the bilateral mPFC, individual structural T1-weighted images were registered to the Montreal Neurological Institute (MNI) space. First, subject-specific T1 images were normalized to MNI space. Based on the Harvard Oxford Atlas, we established the region-of-interest (ROI) for the bilateral mPFC over the falx between both mPFCs. Then, target points in the normalized space were inversely transformed into the original individual space. Based on the CT scan, the sonication path was designed to locate the transducer in the midline, usually perpendicular to the frontal bone, and avoid the frontal sinus (Figure 3).

### 2.6. CT-MRI Coregistration

Acquired CT and MRI scans were spatially co-registered using a 3D slicer (https://slicer.org, accessed on 12 March 2021), and the location of the fiducial markers on CT and MRI was examined to assess the quality of co-registration. Co-registered multimodal images were used in image-guided FUS navigation software.

### 2.7. Electroencephalography (EEG) Recordings

A 19-channel EEG system was recorded using Glass–Telefactor (Beehive Millennium; Glass–Telefactor Corp., Providence, RI, USA). EEGs were recorded at baseline and follow-up visits for 30 min. Pre- and post-stimulation EEGs were recorded over 10 min (5 min eye opened, then 5 min eye opened) before and after sonication. To test the short-term and long-term effects, we recorded post-stimulation EEGs immediately after and 1 week after tFUS, respectively. EEG signals were also recorded over 20 min during sonication without Fp1 and Fp2 channels due to the placement of the transducer on the scalp. The sampling rate was 256 Hz with band filtering at 1 Hz (low cut) and 50 Hz (high cut). Ground and reference electrodes were placed on the right mastoid and ear, respectively.

### 2.8. EEG Analysis

All EEGs were preprocessed on MatLab using the EEGLAB toolbox. The preprocessing procedure was as follows: (1) The recorded EEG was visually examined, and artifact signals and poorly detected channels were removed; (2) automated removal using independent component analysis was applied to eliminate other contaminations, such as eye movement, blinkings, and muscle noise; (3) EEG signals were epoched into 2 s segments. Power spectrum analysis was conducted by filtering into delta (1–3 Hz), theta (4–7 Hz), alpha (8–12 Hz), and beta (13–30 Hz). The procedures of quantitative EEG analysis have been described in detail elsewhere [25]. The relative powers were obtained for each delta, theta, alpha, and beta band and were calculated by dividing the band power by the total power spectral density. Percentage changes were calculated on the basis of the difference of the relative band power between the pre-tFUS and each time point at the post-tFUS values divided by the pre-tFUS value multiplied by 100. To quantify the spectral power change pre- to post-rTMS, the percent change of relative power (*RP*), *R_j_*, at each electrode was defined as:$$R_{j}=\frac{\left(RP_{j}^{post}-RP_{j}^{pre}\right)}{RP_{j}^{pre}}\times 100$$

### 2.9. Statistical Analysis

Statistical analysis was performed using SPSS 19.0 (SPSS Inc., Chicago, IL, USA). The Kruskal–Wallis test for non-parametric method was used to investigate the effect of ultrasound stimulation parameters on the percent change in EEG spectral power at each frequency component.

## 3. Results

### 3.1. Subjects

A total of 11 subjects were recruited from the clinical trial recruiting system, and 3 subjects were excluded (2 subjects had brain neuroimaging abnormalities and 1 subject withdrew consent). A total of 8 subjects with mean age of 58.2 ± 5.6 years (male:female = 3:5, aged between 45 and 65 years) participated in this study. All the subjects were screened and underwent a baseline examination, and then they were allocated into three groups for excitatory, suppressive, and sham stimulations, respectively. The contact surface temperature was 35.4 °C, which is within the safe range according to the official report of the tFUS system (NS-US100, Neurosona Co., Ltd., Seoul, Korea) approved by the Ministry of Food and Drug Safety of Korea. In the previous literature, low-intensity transcranial focused ultrasound (tFUS) was able to alter neural activity without increasing tissue temperature [26]. The tFUS system used in this study has been safely used in other human studies [15]. We also checked whether subjects felt heating from the transducer after every experiment session. No subjects reported any side effects, including heating sense, during and after tFUS, or at follow-up visits. In addition, all the subjects were asked whether they heard any sounds from the transducer during sonication as a possible auditory confound [27,28]. While using the tFUS system NS-US100 in the current study, none of them heard any sounds from the transducer during sonication.

### 3.2. Low-Intensity Focused Ultrasound

The participants were instructed to sit comfortably in armchairs. A fiducial marker was placed on each subject’s head in the same location as during the baseline neuroimaging scan. A pre-stimulation EEG was recorded 10 min before tFUS. The positions of fiducial markers placed on subjects in the tFUS system were registered in subject-specific anatomical neuroimaging, using a probe attached to an optical tracker detected by a motion-tracking infrared camera. The tFUS transducer placed the PVA hydrogel between the tFUS transducer and the scalp as an ultrasonic hydrogel, and then placed it on the front part of the subject’s head. The tracker was attached to the tFUS transducer and the location of the acoustic focus was spatially tracked in the subject’s brain anatomical space. After the acoustic focus was adjusted and positioned to the target brain region, tFUS administration was started. The EEG recording continued during tFUS. Participants were instructed to keep their eyes open and stare at the junction of the walls during the 20 min tFUS stimulation. Subjects were informed every 5 min to avoid them falling asleep during the sonication. The EEG was recorded for 10 min after stimulation and neuroimaging was performed after sonication. During the EEG recording, we visually checked whether there was any noise related to tFUS, but we could not detect any additional tFUS-related noise components.

### 3.3. Targeting of tFUS on the Bilateral Medial Prefrontal Cortex

We used the MNI standard brain anatomy map to stimulate identical brain target regions across all subjects (Figure 3A). First, subject-specific T1 images were normalized to MNI space. The target of the atlas-based ROI of the mPFC was determined in the MNI space. We then inversely transformed the target position in the normalized space from the original individual space. The neurologist examined the target location generated in each subject’s anatomical brain space to ensure that the planned target was well positioned in the mPFC. The target brain regions for tFUS on the brain anatomical MRI are shown in Figure 3B. Through this process, it was confirmed that the MR image-guided ultrasound focus points that stimulated the bilateral mPFC were usually located in the anterior falx between the bilateral mPFC across all subjects.

### 3.4. EEG Activation Changes during and after tFUS

Figure 4 demonstrates scalp maps that show the topographic distributions of the EEG band power change during stimulation and post-stimulation of the mPFC. Spectral power changes compared to the baseline were calculated at four time points with 5 min intervals during 20 min stimulation. Post-stimulation EEG band power changes were estimated at 5 min and 1 week after stimulation to test the short-term and long-term effects of tFUS.

During tFUS, the delta power percent change was lower in the excitatory stimulation parameter relative to the sham stimulation (*p* < 0.001) at 5 min, 10 min, and 15 min after tFUS initiation. The excitatory stimulation showed a decreased percent change in alpha power compared to the suppressive stimulation (*p* < 0.01) at 15 min during stimulation and sham stimulation. During the excitatory stimulation parameter, the beta power percent change was higher at 5 min compared to the suppressive stimulation (*p* < 0.001), and compared to the sham or suppressive stimulations at 20 min (*p* < 0.001). Spectral power percent changes during the suppressive stimulation were lower in the delta band compared to during the sham stimulation (*p* < 0.001) at 10 min and 15 min after sonication initiation, and were greater in the theta (*p* < 0.001) compared to the excitatory stimulation at 10 min after sonication initiation.

In post-stimulation periods, the excitatory stimulation showed a decreased percent change in alpha power compared to the suppressive stimulation (*p* < 0.01) at 5 min and 1 week after stimulation finished compared to the suppressive stimulation. The excitatory stimulation showed that post-stimulation spectral power changes were higher in the beta band compared to the suppressive stimulation (*p* < 0.01) and sham stimulation (*p* < 0.01) at 5 min and 1 week after stimulation finished.

The suppressive stimulation showed a lower percent change in delta power compared to the sham stimulation at 1 week after stimulation (*p* < 0.001), and after 5 min of stimulation, a higher percent change in alpha power was shown compared to the sham (*p* < 0.001).

Figure 5 shows the percent change of the EEG spectral powers post stimulation compared to the pre-stimulation baseline after the excitatory, suppressive, and sham stimulations.

## 4. Discussion

This study used low-intensity tFUS to stimulate bilateral mPFCs simultaneously through a single tFUS session with different parameters, such as excitatory, suppressive, and sham stimuli, in healthy subjects. To the best of our knowledge, our proposed study is the first report using tFUS to stimulate the bilateral mPFC, and to test stimulation effects using different sonication parameters.

The mPFC includes the medial portion of the Brodmann areas (BA) 9–12 and BA 25, and contains reciprocal connections with brain regions that are implicated in emotion processing (amygdala), memory (hippocampus), and higher-order sensory areas (within the temporal cortex). It is also known as one of the core hubs of the DMN, and lesions of the mPFC, which lead to the impairment of emotion regulation, have been related to various neurological and psychiatric disorders, such as depression, anxiety, insomnia, schizophrenia, autism, Alzheimer’s disease, Parkinson’s disease, and addiction [29,30,31]. Therefore, establishing a means of causally regulating mPFC function could be beneficial for studies exploring the evolution of emotional experience and the neural circuits underlying its regulation, and may be helpful for the neuroscience-based treatment of disorders associated with emotional dysregulation.

In this study, we confirmed that changes in EEG spectral power occurred differently depending on stimulation parameters when the bilateral mPFC was stimulated. Of course, it is not yet certain which parameters activate or suppress the brain networks connected to the mPFC; however, it was revealed that brain activation could be modulated through the stimulation of the bilateral mPFC in an experimental setup such as this. In the current study setting using scalp EEG, since the mPFC is located deep in the human brain, it was not possible to confirm that the modulatory effect was localized on the mPFC. In order to address this issue, future studies need to be conducted by using functional neuroimaging analysis. This could be a possible limitation of the current study.

Different tFUS studies in animals and humans have shown that the excitatory or inhibitory effect of tFUS might vary depending on the experimental conditions and stimulation targets. Since the change of sonication parameters may not correspond to a change in the efficacy of the excitatory or inhibitory effect, it is unclear whether ultrasound stimulation with any sonication parameter would have an excitatory or inhibitory effect. In this study, the tFUS parameters were chosen based on previous human ultrasound neuromodulation studies [17]. Sonication times in the previous literature, for instance, varied from 3 to 30 min, and we empirically chose 20 min for sonication time in this study. Optimizing sonication time and other parameters to understand the specific effects of neuromodulation, however, would be needed in further studies. The findings of the present study provided one of the possible sonication parameters of the excitation or suppression of the prefrontal cortex.

With the EEG spectral power change in the excitatory stimulation of tFUS, increased beta power and decreased theta power were found during sonication compared to the suppressive and sham stimulations, and increased beta power lasted until the post-stimulation period. There are several studies reporting that EEG regional power in specific frequency bands is positively or negatively associated with resting-state brain networks, such as the DMN. For example, the resting-state EEG band power at beta frequency was positively correlated with the DMN on an fMRI [32], whereas the theta band power was negatively correlated with the DMN [33]. As specific frequencies of neural activity recorded with EEG are known to be linked to distinct brain functions, increases or decreases in EEG band power after excitatory or suppressive stimulation may reflect the activation or inactivation of brain networks associated with distinct brain functions. In our study, beta band power increased with excitatory stimulation and theta band power increased with suppressive stimulation, compatible with frequency-specific associations between EEG spectral powers and brain networks. Future studies should investigate the association between resting-state brain networks and frequency-specific EEG powers after excitatory and inhibitory tFUS.

As mentioned earlier, tFUS is a promising technique for obtaining the spatially precise neuromodulation of deep-seated brain structures without craniotomy [34]. By focusing on the propagation of acoustic waves through the skull and other tissues, ultrasound stimulation can modulate the dynamics of neuronal activity and induce transient plasticity [4]. The tFUS transducer used in this study can generate a cigar-shaped (approximately 9 mm in diameter and 50 mm in length) sonication focus based on the full width at half-maximum acoustic intensity. Placing the sonication focus of this size in the falx between both mPFCs makes it possible to simultaneously stimulate the bilateral mPFCs (Figure 3B). In addition, we used MNI standard brain anatomical maps to minimize bias in determining stimulation target points. Individual T1-weighted images were normalized in MNI space, and target points identified in the MNI map were inversely transformed into the original images of each participant.

This kind of study setup could show that widespread dynamic changes in brain cortical responses may occur within short periods of time and that the distribution of brain activity changes depends on local brain states and functional brain architecture at the time of tFUS administration and thereafter. Future studies involving larger samples and patients with diseases caused by brain network dysfunction are needed. Moreover, further parameter exploration and safety assessments are needed to provide more convincing evidence of the therapeutic effects of low-intensity tFUS on the mPFC. If so, interventions using this kind of tFUS could be used to alter brain neuronal activity, modulating or reorganizing brain functions in various neuropsychiatric diseases, including sleep disorders and/or depression.

## Figures and Tables

**Figure 1 jcm-11-03809-f001:**
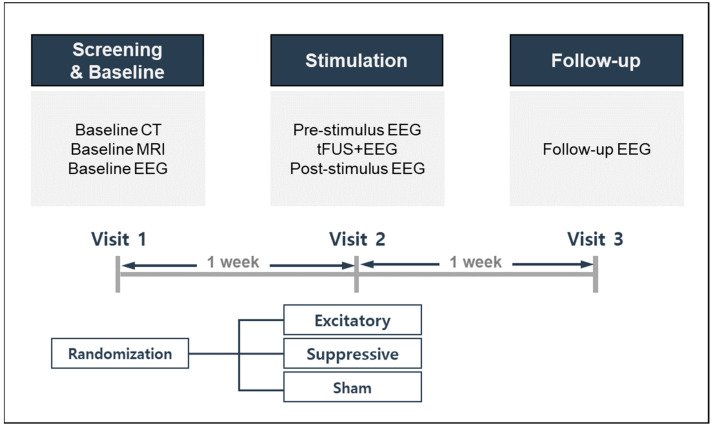
Schematic diagrams of study procedures. CT, computed tomography; MRI, magnetic resonance imaging; EEG, electroencephalography; tFUS, transcranial focused ultrasound.

**Figure 2 jcm-11-03809-f002:**
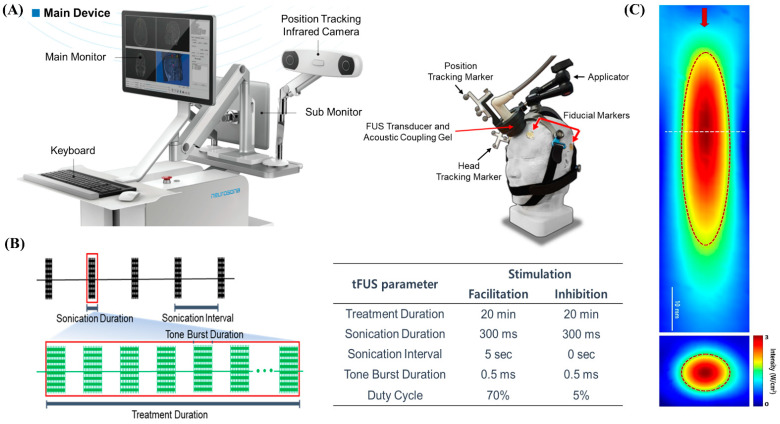
(**A**) Main device of the low-intensity tFUS system. The main device unit consists of a PC and monitor for visualization of neuroimaging and manipulation of tFUS parameters, and an infrared camera for detecting the optical tracker attached to the headgear and the tFUS transducer. The tFUS transducer was placed in front of the subject’s head. The tracker on the transducer was detected by the motion-tracking infrared camera, and the location of the acoustic focus was spatially traced in the subject’s brain anatomical space in real time. (**B**) Schematic diagram of sonication parameters. Excitatory stimulation parameter was a fundamental frequency of 250 kHz, sonication duration of 300 ms, sonication interval of 5 s, tone burst duration of 0.5 ms, and duty cycle of 70%. Suppressive stimulation parameter was a fundamental frequency of 250 kHz, sonication duration of 300 ms, sonication interval of 0 s, tone burst duration of 0.5 ms, and duty cycle of 5%. Sham stimulus was set in the tFUS system. (**C**) Acoustic intensity profile of sonication in longitudinal (YZ) plane along the sonication path and transverse (XY) planes (at the location of the white dotted line) perpendicular to the sonication is shown. The red arrow represents the direction of sonication. The full-width at half-maximum intensity profile is indicated by the dotted red ellipse and circle. Scale bar = 10 mm.

**Figure 3 jcm-11-03809-f003:**
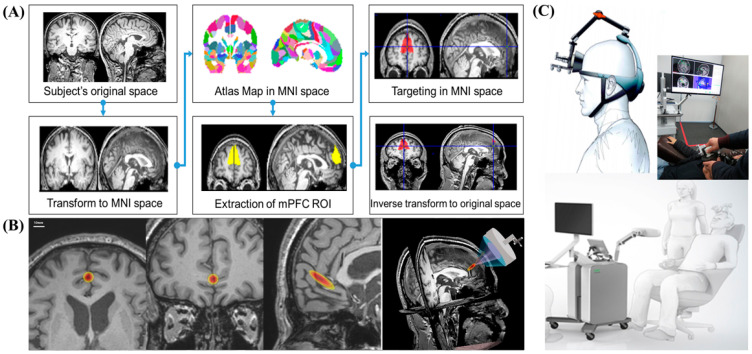
(**A**) MRI-based targeting using a standard MNI brain space. Subject-specific individual T1 structural scans were registered to the Montreal Neurological Institute (MNI) space. Target locations were determined in the individual normalized space by using the atlas-based ROI of mPFC. Then, target points in the normalized space were inversely transformed into the original individual space. (**B**) Targeted brain area and ultrasound field maps for tFUS in the brain anatomical MRI. The planned targets in subject-specific neuroimage data are presented in axial, coronal, sagittal (scale bar = 10 mm), and 3D views. (**C**) Illustration of targeting and sonication of mPFC.

**Figure 4 jcm-11-03809-f004:**
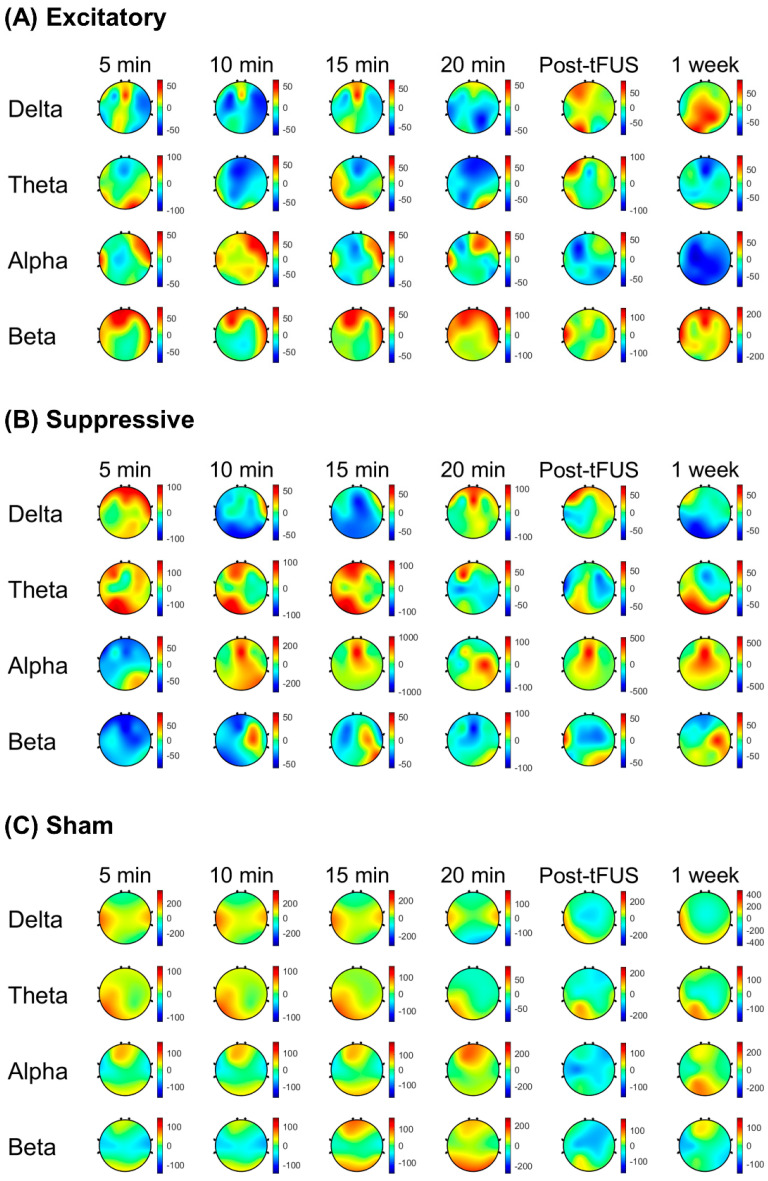
EEG topographic maps averaged their change of spectral powers in response to (**A**) excitatory, (**B**) suppressive, and (**C**) sham ultrasound stimulation at each time point after stimulation initiation across all subjects. Post tFUS: immediately after tFUS finished as short-term effect; 1 week: 1 week after tFUS as long-term effect. During tFUS, delta power percent change was lower in excitatory stimulation parameter relative to sham stimulation (*p* < 0.001) at 5 min, 10 min, and 15 min after tFUS initiation. Excitatory stimulation showed a decreased percent change of alpha power compared to suppressive stimulation (*p* < 0.01) at 15 min during stimulation and sham stimulation. During excitatory stimulation parameter, beta power percent change was higher at 5 min compared to suppressive stimulation (*p* < 0.001), and compared to sham or suppressive stimulation at 20 min (*p* < 0.001). Color bar represents percent change (%) = [each time point after stimulus − pre-stimulation]/pre-stimulation × 100.

**Figure 5 jcm-11-03809-f005:**
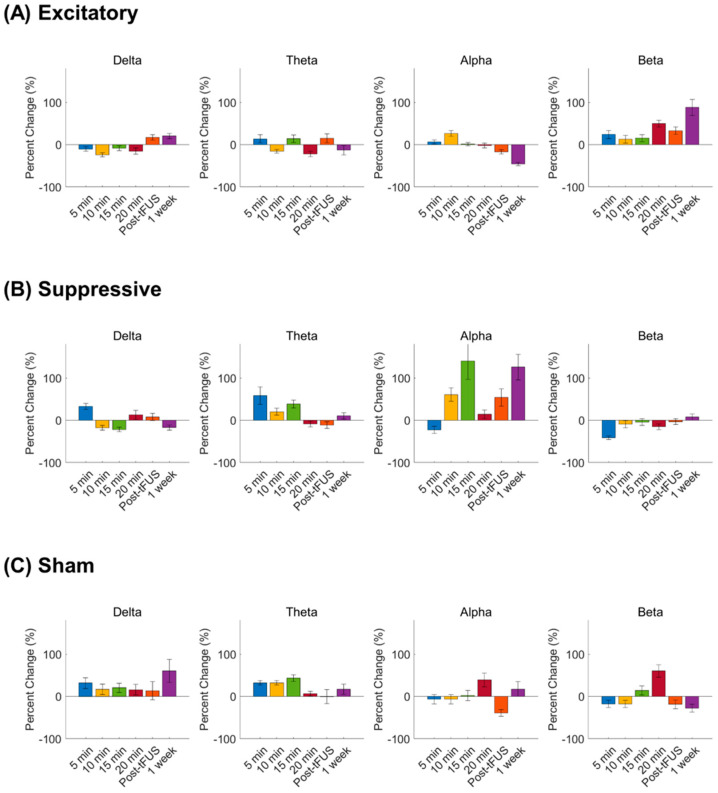
The percent change of EEG spectral powers averaged across all subjects in response to (**A**) excitatory, (**B**) suppressive, and (**C**) sham ultrasound stimulation. Percent change (%) = [each time point after stimulus − pre-stimulation]/pre-stimulation × 100. Bars correspond to the standard error of mean (SEM). Post-tFUS, 5 min after stimulation finished; 1 week, 1 week after stimulation.

## Data Availability

All of the raw and processed data were stored and can be accessed in our laboratory, which is supervised by the corresponding authors, H.W.L.

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
