# Peer review of "Neuromodulation Using Transcranial Focused Ultrasound on the Bilateral Medial Prefrontal Cortex"

_jcm, 2022, doi:10.3390/jcm11133809_

Round 1

Reviewer 1 Report

The authors used ultrasound to modulate the neural activity in mPFC. The Introduction and Methods are well-written. However, some concerns will need to be addressed before the recommendation for acceptance.

Major:

1) What was the intensity or pressure of the ultrasound used in this study?

2) What was the expected percentage of attenuation through the skull?

3) Did the author calculate or measure the temperature increase due to ultrasound sonication? Was it safe for the patients?

4) Recent ultrasonic neuromodulation studies suggested that there might be auditory confounds during sonication. Did the subjects hear any sounds from the transducer? If so, which control experiments performed to eliminate these confounds?

5) Was Figure 4 from two representative subjects or were they averaged maps across all subjects? From the map plot, it is really hard to tell excitatory from suppression. Can the authors elaborate more in the caption to guide the readers how to interpret these plots?

6) Do the authors add Sham plot in figures 4 and 5?

7) Was the modulatory effect localized to mPFC? If so, any evidence to support it?

Minor:

1) Line 17: duplicated ‘Transcranial focused ultrasound (tFUS)’

2) Line 146: ‘70% duty cycle with sonification interval at ms’. Hong long was the interval?

3) The fonts in the color bar of Figure 4 are too small.

Author Response

Response to Reviewer 1 Comments

Dear Editors and Reviewers,

We would like to thank the editors and reviewers for their time and efforts to read our manuscript and for constructive suggestions to improve the work. The reviewer’s comments have been carefully studied and considered in the revision. We have updated the revision with highlights in the manuscript and provided detailed responses for each comment as follows.

Point 1: The authors used ultrasound to modulate the neural activity in mPFC. The Introduction and Methods are well-written. However, some concerns will need to be addressed before the recommendation for acceptance.

What was the intensity or pressure of the ultrasound used in this study?

Response 1. We thank the reviewer for your comment. With the acoustic intensity at focus (AIF) of 3W/cm2 and the fundamental frequency of 250kHz, the acoustic intensity (AI) of 0.30 MPa, mechanical index (MI) 0.60 at the transducer without attenuation, 0.18 MPa, MI 0.36 (attenuation 40%), or 0.15 MPa, MI 0.30 (attenuation 50%) at the focal target. Sonication parameters were described in the revised manuscript on page 4-5, line 155-162 as follows:

“Sonication parameters were as follow: 20 min of 250 kHz tFUS to mPFC with excitatory (70% duty cycle with sonication interval at 5 sec) or suppressive (5% duty cycle with no interval) stimulation protocols. With the acoustic intensity at focus (AIF) of 3W/cm2 and the fundamental frequency of 250kHz, the acoustic intensity (AI) of 0.30 MPa, mechanical index (MI) 0.60 at the transducer without attenuation, 0.18 MPa, MI 0.36 (attenuation 40%), or 0.15 MPa, MI 0.30 (attenuation 50%) at the focal target for both excitatory and suppressive stimulations.”

Point 2: What was the expected percentage of attenuation through the skull?

Response 2. In the previous study (Lee et al., 2016) using the NS-US100 (Neurosona Co., Ltd, Seoul, Korea) same as the tFUS system used in this study, the attenuation of acoustic intensity (AI) was reported to be 58.5±7.3% in the primary somatosensory cortex and 64.1±14.4% in the primary visual cortex. In this study, the attenuation of AI was estimated about 40~50% in the mMFC as indicated above.

References

Lee, W.; Lee, S.D.; Park, M.Y.; Foley, L.; Purcell-Estabrook, E.; Kim, H.; Fischer, K.; Maeng, L.S.; Yoo, S.S. Image-Guided Focused Ultrasound-Mediated Regional Brain Stimulation in Sheep. Ultrasound Med Biol 2016, 42, 459-470, doi:10.1016/j.ultrasmedbio.2015.10.001.

Point 3. Did the author calculate or measure the temperature increase due to ultrasound sonication? Was it safe for the patients?

Response 3. The contact surface temperature is 35.4℃, which is within safe ranges according to the official report of the tFUS system (NS-US100, Neurosona Co., Ltd., Seoul, Korea) approved by the Ministry of Food and Drug Safety of Korea. In the previous literature, low-intensity transcranial focused ultrasound (tFUS) was able to alter neural activity without increasing tissue temperature (Darrow, 2019). The tFUS system used in this study has been safely used in other human studies (Jeong et al., 2021). We also checked whether subjects felt heating from the transducer after every experiment session. No subjects reported any side effects including heating sense during and after tFUS, and at follow-up visits. These have been added in the revised manuscript on page 7, lines 239-247.

Reference

Darrow, D. P. "Focused Ultrasound for Neuromodulation." Neurotherapeutics 2019; 16(1): 88-99.

Jeong, H.; Im, J.J.; Park, J.S.; Na, S.H.; Lee, W.; Yoo, S.S.; Song I.U.; Chung, Y.A. "A pilot clinical study of low-intensity transcranial focused ultrasound in Alzheimer's disease." Ultrasonography 2021; 40(4): 512-519.

Point 4. Recent ultrasonic neuromodulation studies suggested that there might be auditory confounds during sonication. Did the subjects hear any sounds from the transducer? If so, which control experiments performed to eliminate these confounds?

Response 4. We appreciate the reviewer for raising this important point. We acknowledged previous studies reporting auditory confounds during ultrasound sonication (Braun et al., 2020; Sato et al., 2018). Thus, all subjects were asked whether they heard any sounds from the transducer during sonication. Using the tFUS system NS-US100 in this study, there have been no subjects heard any sounds from the transducer during sonication. These have been updated on page 7, lines 247-250.

References

Braun, V.; Blackmore, J.; Cleveland, R.O.; Butler, C.R. Transcranial ultrasound stimulation in humans is associated with an auditory confound that can be effectively masked. Brain Stimulation 2020, 13, 1527-1534, doi:10.1016/j.brs.2020.08.014.

Sato, T.; Shapiro, M.G.; Tsao, D.Y. Ultrasonic Neuromodulation Causes Widespread Cortical Activation via an Indirect Auditory Mechanism. Neuron 2018, 98, 1031-1041 e1035, doi:10.1016/j.neuron.2018.05.009.

Point 5. Was Figure 4 from two representative subjects or were they averaged maps across all subjects? From the map plot, it is really hard to tell excitatory from suppression. Can the authors elaborate more in the caption to guide the readers how to interpret these plots?

Response 5. Figure 4 shows the averaged map across all subjects. To improve the reader’s understanding, we have revised figure 4 and added captions to interpret the plots in the revised manuscript as below on page 8-9, line 285-295: “Figure 4. EEG topographic maps averaged their change of spectral powers in response to excitatory (A), suppressive (B) and sham (C) ultrasound stimulation at each time point after stimulation initiation across all subjects. Post-tFUS: immediately after tFUS finished as a short-term effect; 1 week: one-week after tFUS as a long-term effect. During tFUS, delta power percent change was lower in the excitatory stimulation parameter relative to sham stimulation (P < 0.001) at 5 min, 10 min, 15 min after tFUS initiation. Excitatory stimulation showed decreased percent change of alpha power compared to suppressive stimulation (P < 0.01) at 15 min during stimulation and sham stimulation. During excitatory stimulation parameter, beta power percent change was higher at 5 min compared to suppressive stimulation (P < 0.001), and compared to sham or suppressive stimulation at 20 min (P < 0.001). Color bar represents percent change (%) = [each time point after stimulus – pre-stimulation] / pre-stimulation * 100.”

Point 6. Do the authors add Sham plot in figures 4 and 5?

Response 6.  We have added Sham plot in figures 4 and 5 in the revised manuscript on pages 8 and 10, as the reviewer suggested.

Point 7. Was the modulatory effect localized to mPFC? If so, any evidence to support it?

Response 7. We appreciate the reviewer for this comment. In the current study setting using scalp EEG, it was not possible to confirm that the modulatory effect was localized on mPFC since mPFC is located deep in the human brain. In order to address this issue, further studies need to be conducted by using functional neuroimaging analysis in future studies. This could be a possible limitation of the current study, which was mentioned with another literature review in the Discussion section of the revised manuscript on page 10, line 338-342, and on pages 10-11, line 348-348:

“In the current study setting using scalp EEG, since mPFC is located deep in the human brain, it was not possible to confirm that the modulatory effect was localized on mPFC. In order to address this issue, future studies need to be conducted by using functional neuroimaging analysis. This could be a possible limitation of the current study.

Different tFUS studies from animals and humans showed that the excitatory or inhibitory effect of tFUS might vary depending on experimental conditions and stimulation targets. Since the change of sonication parameters may not correspond to a change of efficacy of excitatory or inhibitory effect, it is unclear whether ultrasound stimulation with any sonication parameter would have an excitatory or inhibitory effect (Zhang et al., 2021). Further studies need to determine sonication parameters for specific effects of neuromodulation. The finding of the present study provides one of the possible sonication parameters to excitation or suppression of prefrontal cortex.”

Reference:

Zhang, T.; Pan, N.; Wang, Y.; Liu, C.; Hu, S. Transcranial focused ultrasound neuromodulation: A review of the excitatory and inhibitory effects on brain activity in human and animals. Front Hum Neurosci 2021, 15, 749162, doi:10.3389/fnhum.2021.749162.

Point 8. Minor: Line 17: duplicated ‘Transcranial focused ultrasound (tFUS)’

Response 8. Thank you for correcting our mistake. We removed the duplicated word as the reviewer commented.

Point 9. Minor: Line 146: ‘70% duty cycle with sonification interval at ms’. Hong long was the interval?

Response 9. The sonication interval was 5 sec, which was missed by mistake. It was a mistake. We apologize for the confusion, and we have corrected it to ‘70% duty cycle with sonication interval at 5 sec’ in the revised manuscript on page 5, line 171-172.

Point 10. Minor: The fonts in the color bar of Figure 4 are too small.

Response 10. We have modified the font in the color bar of Figure 4 on page 8 of the revised manuscript, as the reviewer recommended.

Reviewer 2 Report

The reviewed manuscript describes a neuronavigation-based approach for tFUS to the mPFC. Neuromodulatory techniques than can accurately reach beyond the superficial layers of the cortex are in great need, so this approach is excited and appreciated. There are a few issues with the paper that need to be ironed out before it is suitable for publication.

Major comments

While the authors acknowledge their sample size is low it still remains difficult to ascertain for sure if their techniques reliably modulated EEG. For example, running an ANOVA for sham vs excited means n = 2, and n = 3, yet their F is extraordinarily high for this (54.99), which leads to the question of assumption violation and appropriateness of even running statics over emphasizing the qualitative nature of this project.

In many cases the changes in EEG power are in the same direction for the excitatory and inhibitory tFUS (e.g. delta band), not that this isn’t impossible, but it is counter-intuitive. It would be nice to hear a bit more on why this would be the case.

The authors recorded the EEG during stimulation which could be very interesting but did not mention much of this component during the results. Was there too much noise to analyze this data?

Minor Comments

Some more details on the rational parameters chosen for this study would be helpful. Why was 20min chosen as the sonication time, and why did they choose to image 30min after?

Was the tFUS sized changes per subject? If someone had a particularly large falx the focus beam may not be large enough to innervate into the gray matter.

Author Response

Dear Reviewers,

We would like to thank the reviewers for their time and efforts to read our manuscript and for constructive suggestions to improve the work. The reviewer’s comments have been carefully studied and considered in the revision. We have updated the revision with highlights in the manuscript and provided detailed responses for each comment as follows.

Point 1: The reviewed manuscript describes a neuronavigation-based approach for tFUS to the mPFC. Neuromodulatory techniques than can accurately reach beyond the superficial layers of the cortex are in great need, so this approach is excited and appreciated. There are a few issues with the paper that need to be ironed out before it is suitable for publication.

Major comments: While the authors acknowledge their sample size is low it still remains difficult to ascertain for sure if their techniques reliably modulated EEG. For example, running an ANOVA for sham vs excited means n = 2, and n = 3, yet their F is extraordinarily high for this (54.99), which leads to the question of assumption violation and appropriateness of even running statics over emphasizing the qualitative nature of this project.

Response 1. We appreciate the reviewer for this comment. As you commented, we agree that ANOVA is not a suitable statistical method for testing stimulation effects due to the small sample size. We therefore analyzed the data with the Kruskal-Wallis non-parametric test for comparison of EEG band power among three stimulation conditions. F values are not applicable in the Kruskal-Wallis test. The method for statistical analysis has been updated on page 6, line 227-230, as follows:

“2.9. Statistical analysis

Statistical analysis was performed using SPSS 19.0 (SPSS Inc., Chicago, Illinois, USA). The Kruskal-Wallis test for non-parametric method was used to investigate the effect of ultrasound stimulation parameters on the percent change in EEG spectral power at each frequency component.”

Point 2: In many cases the changes in EEG power are in the same direction for the excitatory and inhibitory tFUS (e.g. delta band), not that this isn’t impossible, but it is counter-intuitive. It would be nice to hear a bit more on why this would be the case.

Response 2. Thank you for your comment. We agree that the decrease in delta band power in both the excitatory and suppressive groups is somewhat counter-intuitive, as the reviewer mentioned. Interestingly, however, excitatory stimulation showed increased beta band power and decreased theta power compared to inhibitory or sham stimulation, reflecting quite opposite effects of these stimulations. The increase in beta band (relatively fast frequency) power appears to be associated with excitatory stimulation, while the increase in theta band (lower frequency) power appears to be associated with suppressive stimulation.

There are several studies reporting that EEG regional power in specific frequency bands is positively or negatively associated with resting-state brain networks, such as default mode network (DMN). For example, the resting-state EEG band power at beta frequency was positively correlated with DMN of fMRI (Neuner et al. 2014), whereas the theta band power was negatively correlated with DMN (Scheeringa et al. 2008). As specific frequencies of neural activity recorded in EEG are known to be linked to distinct brain functions, increases or decreases in EEG band power after excitatory or suppressive stimulation may reflect activation or inactivation of brain networks associated with distinct brain functions. In our study, beta band power increased in excitatory stimulation and theta band power increased in suppressive stimulation, compatible with frequency-specific associations between EEG spectral powers and brain networks. Future studies would investigate the association between resting-state brain networks and frequency-specific EEG power after excitatory and inhibitory tFUS.

We discussed frequency-specific changes of EEG power after excitatory and suppressive tFUS and future directions for further studies in the Discussion section on page 11, lines 359-373.

References:

Neuner, I.; Arrubla, J.; Werner, C.J.; Hitz, K.; Boers, F.; Kawohl, W.; Shah, N.J. The default mode network and EEG regional spectral power: a simultaneous fMRI-EEG study. PLoS One 2014, 9, e88214, doi:10.1371/journal.pone.0088214.

Scheeringa, R.; Bastiaansen, M.C.; Petersson, K.M.; Oostenveld, R.; Norris, D.G.; Hagoort, P. Frontal theta EEG activity correlates negatively with the default mode network in resting state. Int J Psychophysiol 2008, 67, 242-251, doi:10.1016/j.ijpsycho.2007.05.017.

Point 3. The authors recorded the EEG during stimulation which could be very interesting but did not mention much of this component during the results. Was there too much noise to analyze this data?

Response 3. As the reviewer suggested, we have mentioned about the EEG recording during stimulation in the Results section. During the EEG recording, we visually checked whether there was any noise related to tFUS, but we could not detect any additional tFUS-related noise components, which has been added in the revised manuscript on page 7, lines 266-267.

Point 4. Minor Comments: Some more details on the rational parameters chosen for this study would be helpful. Why was 20min chosen as the sonication time, and why did they choose to image 30min after?

Response 4. We appreciate the reviewer for raising this important point. The ultrasound stimulation parameters used in our study were chosen based on previous human ultrasound neuromodulation studies (Zhang, Pan et al. 2021). In the previous literature, sonication times varied from 3 to 30 minutes. We empirically chose 20 min for sonication time, but optimizing sonication time for effective neuromodulation would be needed in future studies. These have been added in the Discussion section of the revised manuscript on page 10-11, line 348-358, as follows:

“Different tFUS studies from animals and humans showed that the excitatory or inhibitory effect of tFUS might vary depending on experimental conditions and stimulation targets. Since the change of sonication parameters may not correspond to a change of efficacy of excitatory or inhibitory effect, it is unclear whether ultrasound stimulation with any sonication parameter would have an excitatory or inhibitory effect. In this study, the tFUS parameters were chosen based on previous human ultrasound neuromodulation studies [17]. Sonication times in the previous literature, for instance, varied from 3 to 30 minutes, and we empirically chose 20 minutes for sonication time in this study. Optimizing sonication time and other parameters for specific effects of neuromodulation, however, would be needed in further studies. The finding of the present study provides one of the possible sonication parameters to excitation or suppression of prefrontal cortex.”

Reference:

Zhang, T., N. Pan, Y. Wang, C. Liu and S. Hu. "Transcranial focused ultrasound neuromodulation: A review of the excitatory and inhibitory effects on brain activity in human and animals. Front Hum Neurosci 2021, 15, 749162, doi:10.3389/fnhum.2021.749162.

Point 5. Was the tFUS sized changes per subject? If someone had a particularly large falx the focus beam may not be large enough to innervate into the gray matter.

Response 5. Thank you for asking such an important question. In this study, we used an image-guided tFUS system with a single-transducer, so as you can see, it is not possible to change the size of the sonication focus on individual subject bases. However, the depth of focus can be adjusted using compressible PVA hydrogels. Therefore, for effective sonication, subjects were initially excluded who had extensive calcification or dilated frontal sinus in the sonication pathway using the baseline neuroimaging exams such as thin-slice CT and T1-MRI. As mentioned in the Material and Methods section, the spatial acoustic intensity profile generated by the tFUS transducer in this study showed that the size of the tFUS focus at full width at half-maximum intensity was approximately 9 mm in diameter and 50 mm in length. And, as shown in the revised Figure 3B, it was confirmed that proper stimulation of bilateral mPFC was possible by targeting the anterior falx through pre-stimulation simulation using the baseline neuroimaging. These have been updated on the revised manuscript on page 4-5, line 155-162.

Round 2

Reviewer 1 Report

Thanks! Looks great!